# Conscientious Objection and Other Motivations for Refusal to Treat in Hastened Death: A Systematic Review

**DOI:** 10.3390/healthcare11152127

**Published:** 2023-07-26

**Authors:** Madalena Martins-Vale, Helena P. Pereira, Sílvia Marina, Miguel Ricou

**Affiliations:** 1Faculty of Medicine, University of Porto, 4200-319 Porto, Portugal; 2CINTESIS@RISE, Faculty of Medicine, University of Porto, 4200-319 Porto, Portugal; up202001140@med.up.pt (H.P.P.); up201410698@med.up.pt (S.M.)

**Keywords:** euthanasia, assisted suicide, hastened death, refusal to treat, conscientious objection, systematic review

## Abstract

Background: Conscientious objection (CO) in the context of health care arises when a health care professional (HCP) refuses to participate in a certain procedure because it is not compatible with their ethical or moral principles. Refusal to treat in health care includes, in addition to CO, other factors that may lead the HCP not to want to participate in a certain procedure. Therefore, we can say that CO is a form of refusal of treatment based on conscience. Hastened death has become an increasingly reality around the world, being a procedure in which not all HCPs are willing to participate. There are several factors that can condition the HCPs’ refusal to treat in this scenario. Methods: With the aim of identifying these factors, we performed a systematic review, following the PRISMA guidelines. On 1 October 2022, we searched for relevant articles on Pubmed, Web of Science and Scopus databases. Results: From an initial search of 693 articles, 12 were included in the final analysis. Several motivations that condition refusal to treat were identified, including legal, technical, social, and CO. Three main motivations for CO were also identified, namely religious, moral/secular, and emotional/psychological motivations. Conclusions: We must adopt an understanding approach respecting the position of each HCP, avoiding judgmental and discriminatory positions, although we must ensure also that patients have access to care. The identification of these motivations may permit solutions that, while protecting the HCPS’ position, may also mitigate potential problems concerning patients’ access to this type of procedure.

## 1. Introduction

Conscientious objection (CO) in the context of health care occurs when a health care professional (HCP) chooses not to perform a certain procedure, considering that it is not in accordance with his/her conscience or morality [1]. It does not necessarily mean that the HCP is against this procedure, but that he/she does not feel morally capable of putting it into practice [1,2,3]. Therefore, we can say that CO is a form of refusal of treatment based on conscience [4].

Refusal to treat in health care is a broader term, and in addition to CO it includes many other motivations. The identification of these motivations and their better comprehension may help to understand which are the areas in which we can act in order to promote better access to care for patients, and to mitigate the impact of treatment. Refusal of treatment can be considered a right, or even a duty, of HCPs, not least because they must act according to the principle of non-maleficence [5,6].

The HCPs’ right to object conscientiously is not a consensual topic [7]. There are two opposing perspectives that can be considered from opposite poles [8]. From a more humanistic or libertarian point of view [8,9,10], we can understand the right to refuse treatment as a way of protecting HCPs from having to perform a procedure against their convictions. Regarding a more professionalist point of view [10,11,12], we can claim that an individual who undertakes to practice a certain profession must be able to assume the values of that same profession as their own and must be willing to perform any procedure that it involves. For this reason, it is important to have CO mediators, in order to protect HCPS’ right of refusal to treat and ensure that this type of position on the part of HPC has a lesser impact on patients. Moreover, it can also be argued that the refusal to perform certain medical procedures by HCPs can place patients in a situation of unequal access, especially in rural and remote areas, where access to health care is not as in-depth [13].

In the health context, there are several situations in which ethical and moral conflicts may arise, which may eventually lead to CO, including contraception, medically assisted procreation, sterilization, genetic tests and prenatal diagnosis [2,14]. Hastened death is also one of the main contexts where HCPs request CO [15]. Its legalization is increasing around the world over the past 20 years [16]. Constituting itself as a controversial procedure in health care since, in addition to being irreversible, it goes against some of the secular assumptions of health care, it is natural that some resistance may arise on the part of HCPs to its implementation [17].

Hastened death is defined as a process that aims to anticipate the death of a person with an incurable and/or fatal disease who demonstrates this explicit desire in a voluntary, conscious and deliberate way [18]. It includes active and voluntary euthanasia (in which a third person, usually a HCP, administers a lethal drug that results in the person’s death) or physician assisted suicide (in which the patient self-administers the lethal drugs prescribed by a physician) [18]. It is a legal practice in a few countries, and those that have made this option available are mostly developed countries [16,19,20].

In these developed countries, there has been an increase in the average life expectancy of the population, and, therefore, a concomitant increase in chronic diseases [21]. These conditions are often associated with long and difficult journeys, which can lead to suffering for patients and their families [22]. Disease-associated pain, suffering, functional and cognitive decline, and associated experiences of loss of dignity and autonomy can motivate hastened death requests [23]. Suffering may result from physical, existential, social, emotional, and spiritual causes, and when it is not relieved can lead the patient to find no other alternative than death to relieve suffering [23,24,25].

Guiding a patient through the process of hastened death is not always an easy task for a HCP, and not everyone has positive experiences with this type of procedure [26,27]. No matter how much the HPC wants to help the patient by alleviating his suffering, there may be reasons that lead the HCP not to want to be involved in hastened death practices, preferring to opt for other end-of-life care options [28].

The existence of conscientious objection assumes that in society there are individuals with different points of view regarding different subjects, and that these must be respected. For this same reason, taking into account the existence of this diversity of perspectives, it is curious that there are cases of hospitals in which all HCPs end up refusing to perform certain controversial practices, such as hastened death. Therefore, the analysis of the reasons that lead HCPs to refuse to carry out this type of practice is very important, as it is a way of being able to create strategies that enable a balance between the protection of the autonomy of HCPs and the right of access to early, safe, and legal healthcare for patients, using hastened death as a reference. For this reason, with this systematic review, we intend to gather existing information on this topic and identify the main factors that condition HCPs’ refusal to treat in hastened death procedures.

## 2. Methods

A systematic review following the PRISMA guidelines [29]. was performed to synthetize the information from studies which explored the factors and motivations behind HPCs’ refusal to treat in hastened death. Primarily, we sought to know if there was any systematic review being performed or recently carried out related to our investigation question: “What are the factors that condition HCPs’ refusal to treat, in the face of hastened death requests?”.

### 2.1. Search Strategy

On 1 October 2022, we used PubMed, Web of Science and Scopus databases to identify the relevant articles, using controlled terms (Medical Subject Headings—MeSH terms) and free text words as main terms: euthanasia (MeSH), assisted suicide (MeSH), assisted suicides (MeSH), physician assisted suicide (MeSH), refusal to treat (MeSH), physician refusal to treat (MeSH) and conscientious objection (Text Word). The search was adapted using the PICo method for systematic reviews (Population, Phenomena of Interest, Context) proposed by the Joanna Briggs Institute (JBI) manual [30]. Search strategies were refined for each database. A detailed description of the research can be found as Appendix A.

### 2.2. Inclusion and Exclusion Criteria

Inclusion criteria: full-text journal articles, performed with HCP or students in the field, without restricting the language or the date of study publication, including all articles related to refusal to treat in a hastened death context. Exclusion criteria: book chapters, dissertations and other similar documents, journal articles not available in full-text or not peer-reviewed. We excluded all the articles that addressed treatment refusal outside the context of hastened death, or those that focused on hastened death without reference to treatment refusals.

### 2.3. Data Extraction and Synthesis

Primarily, we used Mendeley software (Version 1.19.8) to export the references of the selected articles and remove duplicates. In the second place, two reviewers (M.M.V. and H.P.) independently screened the search results, in a first phase by title and abstract, and in a second phase by analyzing the full text of the relevant studies. During this process, uncertainties and doubts that arose were discussed with a third element (M.R.) and resolved by consensus. We excluded articles that did not meet the inclusion criteria.

We extracted information from the selected studies relating to authorship, year of publication, study type, population, sample size, HCP years of professional experience, main factors related to refusals and main motivations related to refusals.

A content analysis of the included studies was made in three steps: skimming, reading, and interpretation of the data [31]. The technique used was categorical analysis by placing our focus on the examination of underlying themes—thematic analysis. After reading the material several times, emergent themes and categories were identified. In detail, from the analysis of each article, we identified the coding units grouped into thematic clusters. Two researchers reviewed all sources identifying core concepts to code the material. Disagreements in the coding process were discussed in the research team.

### 2.4. Quality Assessment

Since we found several types of studies, we chose to perform the quality analyses using the Let Evidence Guide Every New Decision (LEGEND) tool [32], which can be adapted to assess the quality of the different types of articles that we included in the review (qualitative, cross-sectional quantitative, and review or opinion studies). This tool consists of a set of yes or no questions, which vary according to the type of study, and whose joint evaluation allows us to assess essential characteristics in the quality of the articles. Two reviewers independently assessed the quality of each study and resolved discrepancies through discussion with a third element of the research team.

## 3. Results

We identified 693 articles in the initial search. After duplicates removal, 462 articles were screened by title and abstract. Then, 83 were screened based on full text analyses, of which 12 were included in the final study. These results are described in the PRISMA diagram, which provides an overview of studies identified at each stage of the search (Figure 1).

Characteristics of Included Studies: We included 12 studies in this review, of which 8 were qualitative studies, 2 were quantitative studies and 2 opinion articles. Opinion articles were included alongside empirical articles, considering their relevant content to the objectives of the study. The 12 studies were undertaken in Canada (*n* = 6), Australia (*n* = 4), Spain (*n* = 1), and Norway (*n* = 1). Different groups of HCPs were assessed in these different studies, including mostly physicians and nurses, but also pharmacists and medical students. The description of the included studies is presented in Table 1.

In order to summarize the information that we found throughout the various articles, four categories were created that describe treatment refusal: legal, social and technical motivations and CO. Regarding the CO category, it was subdivided into three subcategories, which represent the type of motivations that lead HCPs to opt for CO, including religious, secular, and psychological/emotional motivations. These categories and respective subcategories are summarized in Table 2 and will then be described in more detail in the following paragraphs.

### 3.1. Legal Motivations

We included in this category all those HCPs who claimed not to participate in hastened death procedures because they did not agree with the way the law concerning this was formulated or demonstrated fear of possible legal repercussions arising from the execution of this practice, e.g., in cases where someone close to the patient does not agree with the procedure, and for this reason ends up suing the HCP, putting his professional activity at risk. There were five articles that indicated this factor for refusal of treatment [33,34,35,38,39].

### 3.2. Social Motivations

In this category, we include all individuals who felt that, in some way, the influence of others, whether friends, family, co-workers and bosses, their patients, and even society in general, would have a significant impact on their decision to opt out of hastened death proceedings. These individuals mentioned that the impact of the decision to participate in this type of care could be harmful to their interpersonal relationships, and that it could even undermine their credibility as professionals. Four of the analysed articles demonstrated that this was a factor for treatment refusal [34,35,38,42].

### 3.3. Technical Motivations

This category includes all those individuals whose decision not to participate in this type of procedure is linked to the fact that they do not agree with the way in which it is conducted. Several studies pointed to this factor as something that conditioned the decision of HCPs not to participate in hastened death procedures, either because they did not agree with the drug used in the procedure, because of the type of involvement they would have to have in the process, or because of their preference for other forms of end-of-life care. Six articles pointed to this as a motivating factor for treatment refusal [9,33,34,35,39,43].

### 3.4. Conscientious Objection

CO in health care is defined as the refusal to perform a certain medical procedure based on morality or conscience [1]. In this sense, throughout the various articles [8,30,31,32,33,34,35,36,37,38,39,42], we found different motivations that would lead HCPs to choose to be conscientious objectors, including:Religious/faith motivations

In this category, we include all those individuals who say they choose to object to hastened death, because this procedure is not compatible with the principles of their religion. There were 10 articles whose participants presented this type of justification for being conscientious objectors, and, in fact, there were only two articles that did not refer to this category as a motivation for CO [9,33,34,36,37,38,39,40,41,43].

2.Secular/Moral motivations

In this factor, we included all the arguments relating to the idea that hastened death is not compatible with the basic and traditional principles of medicine. Of the analysed articles, eight of them showed that moral/secular motivations were behind CO [9,34,35,36,37,38,40,41].

3.Psychological/emotional motivations

When we refer to psychological and emotional motivations, we include all those HCPs who ended up preferring to resort to CO, due to the intrinsic feeling of relative discomfort. In fact, these participants referred to feeling intrinsically bad and uncomfortable with the idea of participating in this type of procedure. Six articles included this as one of the motivations for CO [33,34,36,38,39,42].

### 3.5. Quality Analysis of Included Studies

As mentioned, we performed a quality analysis of the included articles, using the LEGEND toll, and the studies were considered of good quality. The quality analysis is presented in detail in the Appendix A.

## 4. Discussion

The objective of this systematic review was to promote better knowledge of the phenomenon of HCP refusal to treat, when faced with hastened death requests.

This study revealed that there are several reasons that may be behind refusal to treat in hastened death procedures, and that they do not necessarily relate to issues of CO, and may be related to other factors, of which legal, technical, and social factors have been highlighted. These categories must be valued since they can still constitute valid reasons for HCPs to choose not to participate in hastened death procedures. CO is related to HPCs’ intrinsic factors, related to their personal characteristics and values, beliefs, and vision of the world [3]. This categorization, that we have found in this systematic review, in addition to allowing a better understanding of the various factors that motivate treatment refusal, also allows us to understand how we can act in various ways, in order to find solutions that respect the decision-making of HPCs, while limiting difficulties in accessing health interventions.

In recent years, there has been a growing trend towards the legalization of hastened death [16,19,20]. Nevertheless, this analysis has shown that there are still many legal factors that motivate refusal to treat by HCP. These factors may be related to their thinking that there are still ambiguities in the law [34], or that it is difficult to define the group of patients to whom these procedures can be applied, or that some HCPs are afraid of suffering legal consequences in cases that the patients’ relatives or other HCPs are not in agreement with the patient’s choice and with the eligibility criteria for hastened death [33,35]. Obviously, hastened death procedures are irreversible, which requires that the legislation around them is made in a clear and unequivocal way, to protect the best interests of patients and HCP [17].

The influence of others can have a great impact on HCPs when it comes to participating in hastened death procedures. Hastened death is often not well accepted socially, especially in rural and smaller communities [38,42]. Older HCPs are also more likely not to accept hastened death procedures [44]. The influence that co-workers and bosses can have on this decision has the potential to be quite impactful, which could help to understand why several services may end up including all professionals as conscientious objectors [45,46]. Many HCPs who work in these environments can end up choosing not to participate in early death procedures. They can fear that this could affect the trust that their patients have in them, and thus harm the doctor–patient relationship [34,35]. HCPs are often concerned that their family members may have a negative opinion regarding their participation in hastened death procedures, and this factor may lead to refusal to treat [35,42].

The technical factors that can motivate refusal to treat can be very comprehensive and may include not agreeing with the type of drug administered, or may have to do with the fact that patients take these drugs autonomously at home, or even with preference for other types of end-of-life care [36,38,39]. All these issues can condition technical difficulties in the application of hastened death procedures. It is not always easy to find a balance between respect for patient’s autonomy and the vulnerability that their illness implies, and in this sense they may end up preferring to provide another type of end-of-life care [17,47]. Furthermore, the way in which the procedure is carried out must be defined by HCPs in a consensual manner, and it must be clear and ensure safety for the patient and for third parties, so that the HCPs feel confident about it.

The boundaries between the motivations for refusal to treat and CO are not always easy to establish. In principle, CO stems from the intrinsic factors of HCPs [3], which can become difficult to distinguish from other motivators, such as those associated with refusal to treat. This distinction is very important, bearing in mind that one of the reasons that legitimizes CO is respect for the individual characteristics of each HCP [1,2]. This no longer applies to other treatment refusal factors, where these factors can more easily be the subject of training and awareness [34].

This analysis suggests a fundamental role for CO in the refusal of treatment by HCPs in hastened death, and this can be motivated by several issues, including religious/faith, secular/moral and emotional/psychological. The importance of these factors is very different from individual to individual and is often related to cultural and educational factors, which condition the personal beliefs and values of each person [48,49].

Secular/moral motivations are mainly linked to the idea that hastened death is a procedure that goes against the basic principles of heath care, such as protection of life and doing no harm [6,50], defended in the Hippocratic oath. Although we can think that this type of motivation can be linked to the fact that HPCs intend to meet the moral and secular principles of their profession, this analysis leads us to believe that HCPs often end up agreeing with their profession’s principles and values, ending up in assuming them as their own [9]. This is the identification that HCPs have with the values and principles of their profession, which makes this a true motivation for CO.

Everyone should have freedom in practicing their religion, and HCPs are no exception [1]. There are health procedures that are not compatible with the principles of certain religions, and hastened death is one of them [51]. Likewise, with this religious freedom, HCPs cannot impose their beliefs on others, and cannot harm their patients due to these beliefs. When a HCP finds himself in a situation in which his religion takes precedence over his professional duty, he can opt for CO [43].

Participation in hastened death procedures can have a negative impact on HCP [52]. To be required to perform a procedure that goes against the convictions of the HCP is often associated with moral distress, which is in turn often associated with negative consequences on health [33,53,54]. Even in cases where HCPs are not against the existence of hastened death procedures, and even if initially they may even have chosen to participate in these procedures, there are reports of HCPs who felt psychologically bad after this participation, having become conscientious objectors [38].

It is important to bear in mind that these factors or motivations do not always appear in isolation and are often interrelated. In addition, the boundary between these various components is often not clearly defined. The difference is not always clear, even for the HCPs themselves, between what is a motivation that leads to CO or to another type of refusal to treat. Therefore, it could be useful to create self-awareness tools that support HCPs in deciding whether to participate in hastened death procedures.

The decision to refuse to treat when a patient asks for it can create a difficult balance between, on the one hand, the protection of professional autonomy and the personal integrity of HCPs, and, on the other hand, the preservation of a non-discriminatory health system with equal access to care for all users [22,55].

Furthermore, the registration of HCPs as conscientious objectors makes it possible to overcome the difficulty of access to hastened death by patients, allowing them to seek out HCPs who are able to perform this procedure [56]. Additionally, this registration as conscientious objectors allows health services to ensure the existence of professionals who are not conscientious objectors [56,57].

The written justification of the reasons why HCPs resort to CO can also be duly justified, because this allows them to self-reflect on their motivations [56]. In addition to functioning as a self-knowledge tool, this justification constitutes a way of avoiding the instrumentalization of CO [57,58].

In cases where HCPs show willingness to explore the psychological dimensions, providing specialized psychological support may help them in managing these negative feelings and emotions [59]. It is also very important that HCPs do not feel alone in the management of hastened death requests. This is one of the reasons why it is important to approach these patients in a collegial manner [60].

It is important to highlight that just because a HCP chooses to refuse treatment, whether for reasons of CO or not, does not mean that he abandons the patient in the process, as it is important that treatment refusal attitudes do not compromise the accessibility of health care [61]. In cases of refusal to treat, referring patients to HPCs who do not have this type of limitation in carrying out the procedure is a way of preventing access from being hindered [62].

Ultimately, the analysis we carried out can allow the creation of self-awareness tools, which can help HCPs to reflect on the different motivations behind refusal to treat in hastened death, and to understand whether they are prepared or not to participate in these procedures.

### 4.1. Strengths and Weaknesses/Limitations of the Study

A strength of this systematic review is the comprehensive search of the three main databases in this investigation area, using adequate terms and with no language and date of publication restrictions. We found that, although there is an increasing number of studies in hastened death and refusal to treat, there is still little scientific evidence regarding the reasons that lead HCPs to refuse to provide this type of care. Although we do not have a very large number of articles, the quality analysis we performed, using an appropriate tool, showed that the articles included in this review were of good quality.

The main limitation of this study is the small number of articles included. Furthermore, this review addresses only part of the phenomenon under study, considering that the studies found are based on self-reports and these may not be the only means of accessing motivations. Therefore, we believe it is important that in the future this type of question is evaluated in more detail, together with HCPs from different areas. Finally, this review focused solely on articles published in the main databases and have not included any sources from the grey literature. However, it might be worth considering conducting a scoping review in the future to explore potential articles that are currently not included in the systematic review.

### 4.2. What This Study Adds

This study highlights several motivations for HCPs’ refusal to treat in a hastened death context, which include CO, but also legal, technical, and social motivations. In addition, three essential motivations for CO in this context are pointed out. We also seek to indicate strategies that promote respect for the individuality of the decisions of each HCP, and that, at the same time, make it possible to mitigate the difficulties of access to this type of procedure for patients.

## 5. Conclusions

There are several factors that condition HCPs’ refusal to treat in the context of hastened death, and the boundaries between them are often not easy to establish. In this regard, it is important to promote the training of HCPs in this topic, as well as to promote reflection on the motivations that lead them not to want to perform this type of procedure. In addition, it is important to promote respect for everyone’s positions regarding hastened death. The decision to participate or not in this type of procedure must be viewed with respect and consideration for the motivations of each professional, avoiding judgmental and discriminatory positions.

HCPs often feel pressured by third parties to participate or not in this type of procedure. It is very important to protect HCPs from being coerced into making such a personal decision as participating in hastened death. Regarding the other categories of refusal to treat, it is important to consider whether they should be accepted, and if the conclusion is that they are legitimate, then these should be formalized.

Finally, it is also important to emphasize the importance of refusal to treat’ mediators in hastened death, to mitigate the problems that may arise in patients’ access to this type of procedure.

## Figures and Tables

**Figure 1 healthcare-11-02127-f001:**
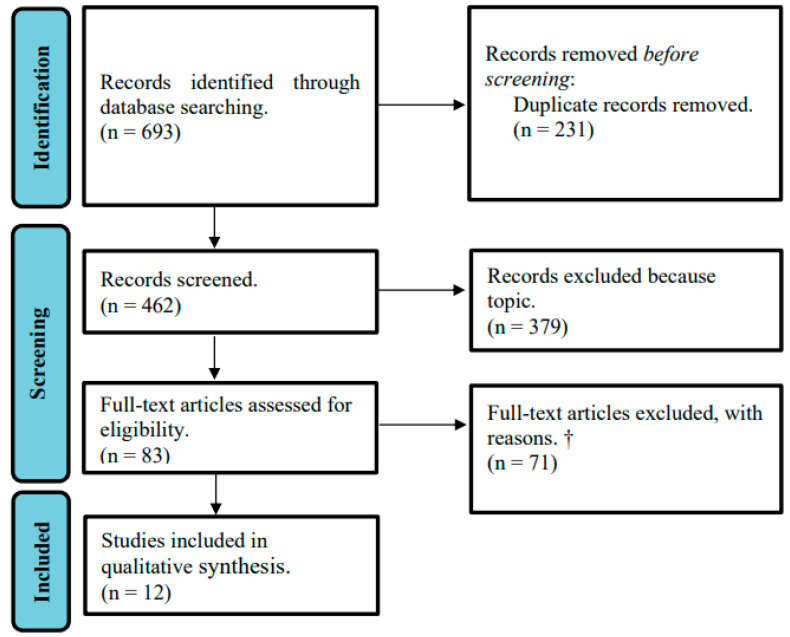
PRISMA 2020 flow diagram for study selection. ^†^ *n* = 34 did not explore the motives that condition HCPs’ refusal to treat, in the face of hastened death requests, but did explore arguments against hastened death; *n* = 19 did not provide information specifically regarding hastened death, but did regarding several medical procedures; *n* = 11 provided arguments against CO in hastened death.; *n* = 7 reflections on the laws that regulate hastened death in various countries.

**Table 1 healthcare-11-02127-t001:** Characteristics of selected studies in this systematic review.

Authorship, Year, and Country (State)	Type of Study	Population	Sample	Main Purpose of Study	Other Motivations for Refusal to Treat	Main Motivations for CO
Beuthin, 2018, Canada [33]	A qualitative study	Nurses	*n* = 17	To understand the range of nurses’ experience in providing care for someone choosing MAiD, whether directly aiding, providing supportive care, or declining to participate.	- Legal factors - Technical factors	- Emotional/Psychological motivations - Religious/Faith motivations
Bouthilier, 2019, Canada (Quebec) [34]	A qualitative study	Physicians that refuse to get involved in their patients’ medical aid in dying requests.	*n* = 22	To explore the CO stated by physicians to understand why some of them refuse to get involved in their patients’ medical aid in dying requests.	- Legal factors - Technical factors - Social factors	- Emotional/Psychological motivations - Religious/Faith motivations - Moral/Secular motivations
Brown, 2021, Canada [35]	A qualitative study	Physicians’ and NPs’ who identified as non-participators in MAiD.	*n* = 35	To identify the factors that influence physicians and NPs when deciding not to participate in the formal MAiD processes of determining a patient’s eligibility for MAiD and providing MAiD and HCPs’ needs in this emerging practice area.	NA	- Emotional/Psychological motivations - Religious/Faith motivations - Moral/Secular motivations
Brown, 2021, Canada [36]	A qualitative study, using an interpretive description methodology	Physicians’ and NPs’ who identified as non-participators in MAiD.	*n* = 35	This research explores the exogenous factors influencing physicians’ and NPs’ non-participation in formal MAiD process.	- Technical factors - Social factor - Legal factors	- Moral/Secular motivations
Carpenter, 2020, Canada (Ontorio) [9]	Opinion study	NA	NA	Discuss main motivations for CO	Technical factors	- Religious/Faith motivations - Moral/Secular motivations
Dumont, 2019, Canada (Quebec) [37]	A qualitative study	Physicians who, in their practice, receive request for medical assistance in dying and who oppose or have a serious reservation about MAID.	*n* = 20	(1) Clarify the ethical issues inherent in physicians’ requests for exemption (2) Present elements for a clear and rigorous position on the conflict between patients’ and physicians’ rights (3) to better understand and describe, based on physicians’ discourse, the reasons (religious and secular) justifying the opposition or reservations of the latter regarding MAID.	NA	- Religious/Faith motivations - Moral/Secular motivations
Haining, 2021, Australia (Victoria) [38]	Qualitative study, phenomenological methodology	Victorian HCPs, in any medical discipline and career stage, who self-identified as having CO to VAD.	*n* = 17	To gain an in-depth understanding of the views of Victorian HCPs, with self-identified CO to VAD, to understand the nature of their CO and the reasons for it.	- Legal factors - Social factors	- Emotional/Psychological motivations - Religious/Faith motivations - Moral/Secular motivations
Isaac, 2019, Australia [39]	A qualitative study	Australian pharmacists	*n* = 14	To investigate Australian pharmacists’ views about their role in PAS, their ethical and legal concerns and overall thoughts about PAS in pharmacy.	- Legal factors - Technical factors	- Emotional/Psychological motivations - Religious/Faith motivations
Nordstrand, 2013, Norway [40]	A quantitative study	Norwegian 5th and 6th year medical student.	*n* = 531	To examine medical students’ views on CO and controversial medical procedures.	NA	- Moral/Secular motivations - Religious/Faith motivations
Pesut, 2019, Canada [41]	Opinion study	NA	NA	Consider the ethical complexity that characterizes nurses’ participation in MAiD and propose strategies to support nurses’ moral reflection and imagination as they seek to make sense of their decisions to participate or not.	NA	- Religious/Faith motivations - Moral/Secular motivations
Pesut, 2020, Canada [42]	A qualitative interview study guided by Interpretive Description using the COREQ checklist	Registered nurses or nurse practitioners who had some experience with participating or choosing not to participate in MAiD.	*n* = 55	Aim and objectives: to describe nurses’ moral experiences with Medical Assistance in Dying in the Canadian context	- Social factors	- Emotional/Psychological motivations
Velasco, 2022, Spain [43]	A quantitative study	All nurses registered in the Official College of Nurses of Madrid.	*n* = 489	To learn about the opinions that the nurses of the autonomous region of Madrid have regarding Euthanasia and Medically Assisted Suicide	- Technical factor	- Religious/Faith motivations

**Table 2 healthcare-11-02127-t002:** Categorization of motivations for CO and other motivations for refusal to treat.

Motivation	Category	Examples Retrieved from Included Studies
Motivations for CO	Religious/faith motivations	“I am an Evangelical Christian, I support life on all levels. And to me this is very personal, that life is a God-given gift and we do not have the right to take that.” [33].
		“I don’t believe that it’s our place to decide when we die; God decides that.” [39].
		“To see someone have a peaceful death and go on their terms, I am happy for them, and I am good with that. But when it comes to if it was me actually administering something to take a life? You know, you kind of think about your own demise. When I get up to the pearly gates, how is that going to be viewed?” [36].
	Emotional/psychological motivations	“It was something really big for me when I saw the death certificate, it was this overwhelming feeling like, oh my gosh, I killed him.” [33].
		“I wake up three o’clock in the morning wondering if I’ve done the right thing, ifF I’ve given the right dose of medication. I don’t want to wake up at three o’clock in the morning knowing that I’ve killed someone … I couldn’t do it” [38].
		“I am in favour of the principle behind medical aid in dying, but I would be very uncomfortable to do it.” [34].
	Moral/Secular motivations	“The existence of these codes strengthens the ability of the profession of medicine to define its moral commitments, even if it sometimes leads to conscientious objections” [9].
		“Until I know everybody will have equal access, I feel that I’m not supportive of the current iteration of voluntary assisted dying.” [38].
		“My job is to make death a positive experience by controlling symptom management. I am not there to bring on death quicker. I am there to support a natural process. The MAID program is not a natural process, it is the exact opposite of what I do.” [36].
Other motivations for refusal to treat	Legal motivations	“The law states that it needs to be the patients who have a prognosis or expected prognosis of six months or less. As doctors we are notoriously bad at predicting prognosis, and we are in a changing landscape where there are lots of advances, so it is very difficult to say when someone is six months or less.” [38].
		“Other HCPs were concerned about the risk of litigation or professional discipline if family members or other HCPs disagreed with the patient’s choice or the HCPs’ eligibility assessments.” [35].
		There are still grey zones in the law. It would help if some criteria were better defined.” [34].
	Technical motivations	“So, it is easy for me to say to patients, “We have to refer you [for formal MAID processes] through the centralized process to the next regional centre.” It is easy for me to say that. So, it gives me a bit of an out.” [35].
		“I’d be worried that the medication will just sit in the cupboard and a partner or child accidentally takes that medicine… As a pharmacist, we see the risks of polypharmacy and stocking of medicines.” [39].
		“More than a third of physicians (36% or 8/22) said they refused due to concerns about their competence, including insufficient clinical experience and being unfamiliar with the pharmaceutical agents used in the protocol: ‘It takes more than a few PowerPoint slides to make you better at this’.” [34].
	Social motivations	I’m a rural and regional (health professional). I walk down the street … I live in the town that I work in and I’ve had people stop me down the street and say, “you’re not going to do that death thing are you?” [38].
		“If I came home at the end of the day and my family asked—‘how was your day Mom?’—would I feel comfortable saying, ‘yeah, I killed someone today’?” [42].
		“Within the (Indigenous) population that I work with, I want to make sure that I am not overstepping my boundaries of trust by being (involved with MAID), or that it would be seen as disrespectful. I do not ever want it to cause distress to the patient.” [35].

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
