# Peer review of "Conscientious Objection and Other Motivations for Refusal to Treat in Hastened Death: A Systematic Review"

_healthcare, 2023, doi:10.3390/healthcare11152127_

Round 1

Reviewer 1 Report

Overall, I believe this is a strong overview of some articles dealing with CO in hastened death, and I congratulate the authors on their thorough and engaging review. It is clear that much time is spend on the review, and interesting elements are presented, especially in the Discussion. Here, especially the trickiness that HCPs find themselves confronted with comes strongly to the fore, and the authors present a nuanced picture on the difficulties these HCPs face. 

The Introduction to the review article is strong and interesting. It clearly states the subject of the paper. However, I do wonder about the background of the authors and the purpose of the paper, and would like to invite the authors to reflect a little bit more on the why of the review. The authors write how "we" need to understand CO motivations, "to promote better access to care or patients." Later in the Introduction, it is stated that the authors aim to "create strategies that enable a balance between the protection of the autonomy of HCPs and the right of access to early, safe, and legal healthcare for patients." But why is this necessary? Can the authors indicate the need for such better access, strategies and balance, e.g., based on literature? And where does this assumption (that access, balance, etc. is needed) come from? Is this something the authors find importance themselves? If so, what then are the authors' backgrounds? Or is this a need also recognized within the academic and/or professional field at large? If so, references can perhaps be added. 

In the methods section, the authors state criteria for inclusion and exclusion of certain academic sources. Can they explain what these criteria are based on (in other words: why are these criteria selected) and what the possible consequences of these criteria might be (or: what other results might the authors have gotten if other criteria were used)? The same questions apply to the choice of the 12 articles: out of the 462 articles that were found, why were these 12 articles chosen specifically? This is especially relevant (in my view) considering the variety between the 12 articles (in terms of country, methods used, etc.). 

I believe the article can be strengthened if in the Discussion section the authors reflect a little bit more on how exactly the gap between HCPs and patients can be bridged. I think it was promised in the Introduction that the authors would give some 'hands-on' ideas, but these aren't explicitly stated enough in the Discussion, in my view. 

There are a few minor mistakes here and there, but nothing a good read-through cannot solve

Author Response

We would like to thank for the reviewer effort and dedicated time to evaluate our manuscript. The points addressed by reviewer allowed us to improve considerably our work.

Reviewer 2 Report

The present review is on a very pertinent topic that needs further research. The review is done, in general, in a scientifically sound way. A few issues should be addressed:

(1) Given the nature of the review, why was grey literature not searched for? This is particularly relevant given the scarcity of studies on the topic and the polemic nature of it. Furthermore, only the main databases were reviewed. This should be mentioned as a limitation alternatively, the authors could consider classifying the study as a scooping review. This might have increased the number of studies included.

(2) The search term “assisted suicides (MeSH)” includes “physician-assisted suicide (MeSH)”. Why include both?

(3) The authors state: “The search was adapted using the PICo method for systematic reviews (Population, Phenomena of Interest, Context) proposed by the Joanna Briggs Institute (JBI) manual.[30] Search strategies were refined to each database.” Perhaps this should be described rather than just mentioned.

(4) Were there no exclusion criteria regarding the targets of the paper? What about hasted death due to an inadvertent side effect of pain management, for example? Were ambiguous cases considered? This should be further explained.

(5) Why were the opinion articles included? This should be fundamental. Why are they included in the analysis alongside empirical studies.

(6) What was the analytical procedure that let to the four categories? Please detail.

(7) I think that the quality of the analysis should be presented in the paper or at least a synthesis of it.

(8) The type of studies reviewed are based on self-reports. Self-reports may not represent in themselves the only means of accessing motivations. This review only addresses part of the phenomena in the study. This should be mentioned.

(9) Was the review pre-registered? Please mention in the paper

Good enough english

Author Response

We would like to thank for the reviewer effort and dedicated time to evaluate our manuscript. The points addressed by reviewer allowed us to improve considerably our work. Please see the attachment.
